# Accelerated Evolution of Cytochrome *c* in Higher Primates, and Regulation of the Reaction between Cytochrome *c* and Cytochrome Oxidase by Phosphorylation

**DOI:** 10.3390/cells11244014

**Published:** 2022-12-12

**Authors:** Sue Ellen Brand, Martha Scharlau, Lois Geren, Marissa Hendrix, Clayre Parson, Tyler Elmendorf, Earl Neel, Kaila Pianalto, Jennifer Silva-Nash, Bill Durham, Francis Millett

**Affiliations:** 1North Murray High School, Chatsworth, GA 30705, USA; 2Department of Chemistry and Biochemistry, University of Arkansas, Fayetteville, AR 72701, USA; 3Independent Researcher, P.O. Box 603, Dardanelle, AR 72834, USA; 4School of Medicine, University of Kansas Medical Center, 2060 W 39th Ave, Kansas City, KS 66103, USA; 5Tulsa Bone and Joint Associates, Tulsa, OK 74146, USA; 6Independent Researcher, 5 Kingdom Court, Maumelle, AR 72113, USA

**Keywords:** Cytochrome *c*, Cytochrome *c* oxidase, electron transfer, phosphorylation

## Abstract

Cytochrome *c* (Cc) underwent accelerated evolution from the stem of the anthropoid primates to humans. Of the 11 amino acid changes that occurred from horse Cc to human Cc, five were at Cc residues near the binding site of the Cc:CcO complex. Single-point mutants of horse and human Cc were made at each of these positions. The Cc:CcO dissociation constant K_D_ of the horse mutants decreased in the order: T89E > native horse Cc > V11I Cc > Q12M > D50A > A83V > native human. The largest effect was observed for the mutants at residue 50, where the horse Cc D50A mutant decreased K_D_ from 28.4 to 11.8 μM, and the human Cc A50D increased K_D_ from 4.7 to 15.7 μM. To investigate the role of Cc phosphorylation in regulating the reaction with CcO, phosphomimetic human Cc mutants were prepared. The Cc T28E, S47E, and Y48E mutants increased the dissociation rate constant k_d_, decreased the formation rate constant k_f_, and increased the equilibrium dissociation constant K_D_ of the Cc:CcO complex. These studies indicate that phosphorylation of these residues plays an important role in regulating mitochondrial electron transport and membrane potential ΔΨ.

## 1. Introduction

Cytochrome *c* (Cc) is a very ancient protein occurring first in anaerobic photosynthetic bacteria over 3 billion years ago, possibly resembling cytochrome *c*_2_ (Cc_2_) in the current photosynthetic bacterium *Rhodobacter sphaeroides* [1]. Cc is a very highly conserved protein with the heme iron liganded by a His nitrogen and a Met sulfur (Figure 1) [2]. Extensive studies have identified a highly conserved ring of positively charged lysines surrounding the heme crevice of Cc that form electrostatic interactions with negatively charged Asp and Glu residues at the binding site of the Cc partners (Figure 1). Chemical modification, mutagenesis, and X-ray crystallography studies have shown that the *Rb. sphaeroides* Cc_2_—reaction center complex is stabilized by electrostatic interactions between the ring of lysines surrounding the heme crevice of cyt c_2_ and Asp and Glu residues on the reaction center [3,4,5]. The rate constant for electron transfer from cyt c_2_ to the photooxidized chlorophyl dimer D^+^ was k_et_ = 1.1 × 10^6^ s^−1^ in both the crystal complex and the complex in solution, demonstrating that the structure of the complex was the same in the crystal and in solution [5]. The experimental rate constant of k_et_ = 1.1 × 10^6^ s^−1^ was in good agreement with calculations based on Marcus theory using the closest distance R = 14.2 Å between the conjugated ring atoms of the cyt c_2_ heme and the chlorophyl dimer D^+^ [5].

Chemical modification studies demonstrated that lysines 8, 13, 27, 72, 79, and 87 surrounding the heme crevice of horse Cc are involved in binding negatively charged residues on cytochrome b_5_, in agreement with a computational model for the Cc–cyt b_5_ complex [6,7]. A new laser-activated ruthenium technique was developed to measure the rate constant for intracomplex electron transfer between cyt b_5_ labeled with a ruthenium complex at Cys-65 (Ru-65-Cyt b_5_) and Cc to be 4 × 10^5^ s^−1^, which is in good agreement with theoretical predictions of the rate constant [8,9].

Chemical modification studies demonstrated that the ring of lysines surrounding the heme crevice of Cc is involved in binding negatively charged residues on yeast cytochrome c peroxidase [10,11], in agreement with an X-ray crystal structure of the yeast Cc–cytochrome c peroxidase complex [12]. A new ruthenium-labeled yeast Cc derivative, Ru-39-Cc was designed to measure the actual rate of intracomplex electron transfer to the Trp-191 radical cation and the oxyferryl heme in cytochrome c peroxidase compound I [13]. Photo-reduced Ru-39-Cc initially transferred an electron to the radical cation in CMPI with rate constant k_eta_ = 2 × 10^6^ s^−1^, and then reduced the oxyferryl heme with rate constant k_etb_ = 5000 s^−1^. Calculations based on Marcus theory using the distance R = 16 Å between the Ru-39-Cc heme and the Trp-191 radical cation in the X-ray crystal structure gave a rate constant of k_etb_ = 3 × 10^5^ s^−1^, somewhat less than the experimental rate constant [13].

Extensive chemical modification studies have demonstrated that lysines 8, 13, 27, 72, 79, 86, and 87 surrounding the heme crevice of Cc are involved in binding negatively charged residues on both cytochrome *bc*_1_ (cyt bc_1_) and cytochrome *c* oxidase (CcO) in the mitochondrial respiratory chain, indicated that Cc must first bind to cyt bc_1_ and accept an electron, then be released, bind to CcO and transfer an electron to CcO [14,15,16,17,18,19,20,21,22,23,24,25]. Lang and Hunte [26,27] have determined an X-ray crystal structure of the yeast Cc–cyt bc_1_ complex that is in agreement with the chemical modification studies. Laser excitation of a 1:1 complex between reduced yeast Ru-39-Cc and yeast cyt bc_1_ at low ionic strength leads to rapid reduction of heme c, followed by intracomplex electron transfer from cyt c_1_ to heme c with a rate constant of 77,000 s^−1^ [28,29]. Mutation of Arg-13 and lysines, 27, 72, 79, 86, and 87 to Ala in yeast Ru-39-Cc resulted in significant decreases in the rate of complex formation with cyt bc_1_, in agreement with the crystal structure [28]. There is a Π–cation interaction between Cc Arg-13 and cyt c_1_ Phe-230 which has been shown to play an important role in the interaction between the two proteins [27]. Intracomplex electron transfer from laser excited horse Ru-72-Cc to either bovine or *Rb. Sphaeroides* cyt bc_1_ heme c_1_ had a rate constant of 60,000 s^−1^ [30]. Mutagenesis studies of the acidic residues Glu-74, Glu-101, Asp-102, Glu-104, Asp-109, Glu-162, Glu-163, and Glu-168 surrounding the heme crevice of *Rb. Sphaeroides* cyt c_1_ demonstrated they are involved in binding Cc [30], in agreement with the X-ray crystal structure of the yeast Cc–cyt bc_1_ complex [27]. Calculations based on Marcus theory using the distance R = 9.4 Ǻ between the Ru-39-Cc heme and the cyt c_1_ heme in the X-ray crystal structure gave a rate constant of k_etb_ = 1.8 × 10^5^ s^−1^, somewhat larger than the experimental rate constant [30].

CcO is the terminal protein in the electron transport chain of mitochondria as well as many bacteria [31,32]. Cc binds to subunit II of CcO and transfers an electron to Cu_A_ [33,34,35,36,37,38]. Electrons are then transferred sequentially from Cu_A_ to heme a, heme a_3_ and Cu_B_ in CcO subunit I, reducing oxygen to water and generating a proton gradient across the membrane. The highly conserved ring of seven lysines surrounding the Cc heme crevice have been shown by chemical modification studies to be involved in electrostatic interactions with CcO [14,15,16,17,18,22,39]. The X-ray crystal structures of bovine [40], *Paracoccus denitrificans* [41], and *Rb. sphaeroides* CcO [42,43] have shown a central hydrophobic domain surrounded by a prominent ring of acidic residues on subunit II involved in the binding of Cc [44,45,46,47,48,49,50,51]. Computational studies by Roberts and Pique [52] indicate that the interaction between horse Cc and bovine CcO involves a central hydrophobic domain surrounded by electrostatic and hydrogen bonding interactions (Figure 2). Laser excitation of the 1:1 complex between horse Ru-39-Cc and bovine CcO at low ionic strength led to electron transfer from heme c to Cu_A_ with rate constant k_3_ = 6 × 10^4^ s^−1^, followed by electron transfer from Cu_A_ to heme a with rate constant k_5_ = 1.8 s 10^4^ s^−1^ [34]. The ruthenium complex is on the backside of Ru-39-Cc, and does not affect any of the steady-state kinetic parameters of the reaction with CcO [34] or cyt bc_1_ [29]. The K13A, K72A, K86A, and K87A Ru-39-Cc mutants greatly decreased the rate constant k_2nd_ for binding Ru-Cc to CcO at 150 mM ionic strength, indicating that lysines 13, 72, 86, and 87 are involved in electrostatic binding to CcO (Figure 2) [53]. However, the K13A, K72A, K86A, and K87A Ru-39-Cc mutants had nearly the same intracomplex rate constant k_3_ as wild-type Ru-39-Cc, demonstrating that these lysines are not involved in the electron transfer pathway (Figure 2) [53]. The *Rb. sphaeroides* (*Rs*) CcO mutants E148Q, E157Q, D195N, and D214 N significantly decreased the second-order rate constant k_2nd_ for the reaction with horse Ru-55-Cc, indicating that the negative charges on E148, E157, D195, and D214 (homologous to bovine CcO E109, D119, D139, and D158) are involved in binding Cc [48]. The *Rb. sphaeroides* W143F mutant (bovine W104) decreased k_3_ by 450-fold but did not affect the dissociation constant K_D_ of the Cc:CcO complex or the redox potential of Cu_A_ [48]. These results are consistent with the computational model for the Cc:CcO complex with the electron transfer pathway: heme c → CcO-W104 → CcO-M207 → Cu_A_, which has a Van-der-Waals interaction between heme c CBC methyl group and the W104 C-H bond, a separation of 2.8 Å between the W104 CH and the sulfur atom of Met-207, and 9 covalent bonds (Figure 2) [52]. Calculations based on Marcus theory using the distance R = 14.1 Å between the Ru-39-Cc heme and the Cu_A_ center in the computational complex give a rate constant of k_3_ = 5 × 10^4^ s^−1^, in good agreement with the experimental rate constant k_3_ = 6 × 10^4^ s^−1^ [53]. The 450-fold decrease in k_3_ caused by the *Rb. sphaeroides* CcO W143F mutation [48] is consistent with an additional 2.4 Å gap between the Cc heme CBC methyl group and CcO F143 [53].

A crystal structure for the Cc–CcO complex [54] is very different from the computational model of the complex [52]. The binding domain of the Cc-CcO crystal structure is quite small, with several interactions of Cc residues K8, Q12, K13, and K87 with CcO residues. Calculations based on Marcus theory using the distance R = 18.7 Å between the Ru-39-Cc heme and the Cu_A_ center in the crystal structure give a rate constant of k_3_ = 90 s^−1^, much smaller than the experimental rate constant k_3_ = 6 × 10^4^ s^−1^ [53]. A new pathway for electron transfer from Cc heme to Cu_A_ in the Cc:CcO crystal structure was proposed:heme c → Cc-C14 → Cc-K13 → CcO-Y105 → CcO-M207 → Cu_A_.

This is a long pathway with a running distance of 41.9 Å [54], with a calculated rate constant of k_et_ = 3.3 × 10^−5^ s^−1^ using the Beratan tunneling pathway theory [53]. The Ru-39-Cc K13A mutant had very little effect on the experimental rate constant k_3_ for electron transfer from Cc heme c to Cu_A_, indicating that Cc Lys 13 is not on the electron transfer pathway [53]. The Rs CcO subunit II Y144 (bovine Y105) mutation caused only a 3.8-fold decrease in k_3_, indicating that Y105 is also not on the electron transfer pathway [53]. The Cc:CcO crystal structure [54] is therefore unlikely to be active in electron transfer. The configuration of the Cc:CcO crystals could have been affected by crystal packing forces or by the conditions used in crystallization. 

Cc is one of the most highly conserved proteins and is an excellent marker for evolutionary changes in species over the history of life on earth. Cc underwent three periods of accelerated evolution, the first early in vertebrate evolution, the second at the stem of the anthropoid primates, and the third in the catarrhine stem leading to the Old-World monkeys, apes, and humans [55]. Of the 11 amino acid changes that occurred from horse Cc to human Cc, eight occurred in the lineage from anthropoid primates to humans [55]. Five of these changes from horse to human Cc occurred at residues near the binding site of the Cc:CcO complex, V11I, M12Q, D50A, A83V, and T89E (Figure 1 and Figure 3). The other six changes occurred farther from the binding site, and may also play a role in the function of Cc: A15S, F47Y, T58I, K70G, E62D, and E92A. There are only two amino acid changes from horse to bovine Cc, K60G and T89G [2], which should not affect the function of Cc. The binding site on CcO for Cc underwent a parallel accelerated evolution in the lineage from anthropoid primates to humans [56]. The CcO residues near the Cc binding site had over four times the number of changes as residues in other regions. It has been suggested that the rapid co-evolution of Cc and CcO has resulted in a very specific interaction between human Cc and human CcO [57]. One plausible reason for the accelerated evolution of Cc and CcO in primates is the greater need for energy production to support the expansion of the neocortical portion of the brain [58]. Another reason is the need to regulate the rate of mitochondrial electron transfer and membrane potential ΔΨ to minimize formation of reactive oxygen species (ROS) and support the longer lifespan of humans [59]. In the present studies, the effect of single-point mutations at residues 11, 12, 50, 83 and 89 in horse and human Cc on electron transfer and binding Cc to CcO was determined. It was found that the dissociation constant K_D_ of the Cc:CcO complex was much smaller for human Cc than for horse Cc, indicating much tighter binding for human Cc. In the human mutants, K_D_ decreased in the order: native horse Cc > human A50D > V83A > I11V > N. Human Cc > M12Q > E89T. Corresponding results were found for horse Cc mutants.

The reaction between Cc and CcO plays a key role in regulating electron transfer and energy production in the entire mitochondrial respiratory chain [60]. Phosphorylation of key serine, threonine, and tyrosine amino acid residues on Cc and CcO has been found to regulate mitochondrial electron transfer and membrane potential ΔΨ, and minimize formation of ROS [59,60,61,62,63]. Defects in regulation of the reaction between Cc and CcO have been linked to many degenerative diseases and aging, and also to reperfusion injury following stroke and heart attack [60]. Cc is also the key activator of apoptosis, which involves release of dephosphorylated Cc from the mitochondria to the cytoplasm, and formation of the apoptosome, leading to cell death [59]. Phosphorylation of Cc minimizes release from the mitochondria and activation of apoptosis [59]. Cc can be phosphorylated at five different residues, Tyr97, Tyr48, Thr28, Ser47, and Thr58 in different tissues of the body [59,62,63]. In the present studies, the following phosphomimetic human Cc mutants have been prepared: Cc T28E, S47E, Y48E, Y97E, and the double mutant S47E-Y48E. The negatively charged carboxyl group on the glutamate mimics the phosphorylated serine, threonine or tyrosine. Laser kinetic studies indicated that the Cc T28E, S47E, and Y48E mutants increased the dissociation rate constant k_d_ of the Cc:CcO complex and decreased the formation rate constant k_f_. Ultracentrifuge studies indicated that the equilibrium dissociation constant K_D_ of the Cc:CcO complex was increased by the Cc T28E, S47E, and Y48E mutants. These studies demonstrate that introduction of a negative charge at Cc T28, S47, or Y48 decreases the strength of the interaction between Cc and CcO, and indicates the phosphorylation of these residues plays an important role in regulating mitochondrial electron transport and membrane potential ΔΨ.

## 2. Materials and Methods

### 2.1. Materials

Horse heart cytochrome c (Type VI), lauryl maltoside, carboxyl-2,2,5,5-tetramethyl-1-pyrolidinyloxy free radical (3CP), and aniline were obtained from Sigma-Aldrich, Inc., P.O. Box 14508, St. Louis, MO 68178, United States. The horse cytochrome c mutant, K39C Cc was prepared as described by Engstrom et al. [29]. Bovine CcO was prepared as described by Pan et al. [33]. 

### 2.2. Site-Directed Mutagenesis

The native horse Cc and human Cc plasmids were a generous gift from Gary Pielak and are described in Olteanu et al. [64,65]. Single point mutations in native human or horse plasmid were made using the Strategene QuikChange Site-Directed Mutagenesis Kit. The oligonucleotides for the horse mutations (D50A, V11I, A83V, T89E, and Q12M) and human mutations (A50D, I11V, V83A, E89T, M12Q, T28E, S47E, Y48E, and Y97E) were synthesized by Integrated DNA Technologies, Coralville, IA. Native and mutant Cc plasmids were sequenced by the University of Arkansas DNA resource center, Fayetteville, AR. Expression and purification procedures were followed as described by Olteanu et al. [64] except a waters 600 E HPLC and waters 991 photodiode array detector was used. To confirm the total mass and mass of digested protein fragments, MALDI TOF mass spectrometry was done on all mutants at the University of Arkansas Statewide Mass Spectrometry Facility, Fayetteville, AR. 

### 2.3. Analytical Ultracentrifuge Experiments

Solutions for ultracentrifuge experiments contained 5 μM native or mutant Cc, 6 μM bovine CcO, 0.2% lauryl maltoside, and 5 mM phosphate buffer pH 7.0 and 0–350 mM NaCl. The sample and blank were injected into a double sector cell assembly and placed in an AN-60Ti rotor. The sample was spun at 48,000 r.p.m. and 22 °C using a Beckman XL-A analytical ultracentrifuge, and the 410 nm absorbance was measured every 10 min for a total time of 75 min. The sedimentation velocities are unbound Cc: 1.74 S, unbound CcO: 9.7 S and the 1:1 Cc:CcO complex: 11.4 S [48]. The ultracentrifuge experiments were analyzed assuming that oxidized Ccforms a 1:1 complex with CcO, which has been previously verified under these conditions:(1)$$\begin{array}{l} \;\;\;\;\;\;\;\cdot \cdot \cdot \cdot \cdot \cdot k_{f}\\ \mathrm{Cc}+\mathrm{CcO}⇄\mathrm{Cc}:\mathrm{CcO}\cdot \cdot \cdot \cdot K_{D}=k_{d}/k_{f}=[\mathrm{Cc}][\mathrm{CcO}]/[\mathrm{Cc}:\mathrm{CcO}]\\ \;\;\;\;\;\;\;\cdot \cdot \cdot \cdot \cdot \cdot k_{d} \end{array}$$

The concentration of free Cc, [Cc], was calculated from the absorbance in the free Cc plateau region using an extinction coefficient of 106 mM^−1^ cm^−1^ at 410 nm. The concentration of the 1:1 Cc:CcO complex was [Cc:CcO] = [Cc_T_] − [Cc], and [CcO] = [CcO_T_] − [CcO], where [Cc_T_] and [CcO_T_] are the total concentrations of Cc and CcO, respectively. Following each run, the concentration of salt within the sample cell was increased by adding 1 M NaCl. The salt concentrations for each ultracentrifuge run were 0, 70, 100, 120, 150, and 350 mM. The final salt concentration (350 mM) was used to completely inhibit Cc binding to CcO.

### 2.4. Flash Photolysis Experiments

Laser flash photolysis experiments involving Ru-39-Cc were carried out as previously described [34,48,53]. The reactions of Cc heme, CcO CuA, and CcO heme a were monitored at 550 nm, 830 nm, and 605 nm, respectively [34,48,53,66,67,68]. 

Reactions were carried out in 2 mM sodium phosphate, pH 7.0 or 5 mM TrisCl, pH 8.0 with 0.1 % dodecyl maltoside at 22 °C. 10 mM aniline and 1 mM 3CP were included as sacrificial electron donors D to reduce Ru^III^ to Ru^II^ (Figure 1). Numerical integration was used to fit the time dependence of the absorbances to Figure 2 [53]. 

At low ionic strength when CcO was in excess of Ru-39-Cc, only a single phase of Ru-39-Cc oxidation was observed, which is due to intracomplex electron transfer from photo-reduced heme c to CcO Cu_A_ in the 1:1 Ru-39-Cc:CcO complex. As the ionic strength was increased, a second slow phase with rate constant k_obs_ is observed due the second-order reaction of free Ru-39-Cc^r^ with CcO. At intermediate ionic strength, k_f_, k_d_, and K_D_ could be measured from the fraction of the fast phase and the rate constant of the slow second-order phase as shown below [34]. Previous studies have established that k_3_ >> k_d_ [34]. Therefore, the rate equation below is valid for the slow second-order reaction k_obs_:d[Cc^r^]/dt = −k_f_ [CcO][Cc^r^] = −(k_f_[CcO])[Cc^r^] = −k_obs_ [Cc_r_] k_obs_ = k_f_[CcO]···k_f_ = k_obs_/[CcO](2)

The fraction of the fast phase is f = ΔA_fast_/(ΔA_fast_ + ΔA_slow_)
    [Cc_T_] and [CcO_T_] are the total concentrations of Cc and CcO.
     [Cc] = (1 − f)[Cc_T_]······[Cc:CcO] = f [Cc_T_] [CcO] = [CcO_T_] − f [Cc_T_]
K_D_ = [Cc][CcO]/[Cc:CcO] = ((1 − f)([CcO_T_] − f[Cc_T_]))/f(3)
k_f_ = k_obs_/([CcO_T_] − f [Cc_T_])(4)
k_d_ = k_f_ K_D_ = k_obs_ (1 − f)/f(5)

At high ionic strength where K_D_ >> [CcO_T_], and f = 0
k_f_ = k_obs_/[CcO_T_](6)

At low ionic strength where K_D_ << [CcO_T_], a slow second-order reaction can only be observed when [Cc_T_] > [CcO_T_]. Under these conditions, f = [CcO_T_]/[Cc_T_] and
k_d_ = k_obs_([Cc_T_] − [CcO_T_])/[CcO_T_](7)

Experiments were also carried out to study the kinetics of the reaction between Cc and CcO using laser-excited lumiflavin to rapidly transfer an electron to unbound oxidized Cc to form Cc^r^ [34,69]. Excited lumiflavin cannot transfer an electron to Cc in the Cc:CcO complex because the heme crevice of Cc is not accessible. Excited lumiflavin also cannot reduce CcO Cu_A_ or heme a [69]. The lumiflavin flash photolysis technique is limited to the measurement of rate constants k_obs_ of less than 2000 s^−1^. It was used to measure k_d_ at low ionic strength, and k_f_ at high ionic strength. 

## 3. Results

### 3.1. Effect of Mutations from Horse Cc to Human Cc on the Reaction with CcO

The role of amino acid residues that are mutated from horse Cc to human Cc was studied by preparing horse Cc mutants in which one of the Cc residues V11, Q12, D50, A83, and T89 was changed to the amino acid in human Cc. A similar set of human Cc mutants was prepared. 

The effect of the horse and human Cc mutants on the dissociation constant K_D_ of the 1:1 Cc:CcO complex was measured using analytical ultracentrifuge sedimentation velocity experiments of 5 μM Cc and 6 μM CcO in 5 mM sodium phosphate pH 7.0 and 0–300 mM NaCl, as previously described [48]. At ionic strengths below 50 mM, the K_D_ values of native horse and human Cc were <0.2 μM, too small to measure, indicating very strong binding. At ionic strengths greater than 150 mM, the K_D_ values were >30 μM, indicating very weak binding. The K_D_ values of horse and human Cc and all the mutants at an ionic strength of 110 mM are shown in Table 1. Native human Cc has a K_D_ value of 4.7 μM, compared to 28.4 μM for native horse Cc, indicating much stronger binding of human Cc to bovine CcO. The largest effect of the mutations was at residue 50, where the A50D human Cc mutant increased K_D_ to 15.7 μM, and the D50A horse Cc mutant decreased K_D_ to 11.8 μM. The E89T human Cc mutant decreased K_D_ to 2.1 μM, while the T89E horse Cc mutant increased K_D_ to >35 μM, indicating that the negative charge on E89 decreases binding strength.

K_d_ was measured by an analytical ultracentrifuge using 5 μM native or mutant Cc, 6 μM native Bovine CcO in 5 mM Pi pH 7.0, 100 mM NaCl, 0.2% Lauryl maltoside. The error limits are ±20%.

Horse, human, and human A50D Ru-39-Cc mutants were prepared to study the rapid kinetics of Ru-39-Cc with CcO. Laser excited Ru^II^* transfers an electron to heme c Fe^III^ in Ru-39-Cc to form Ru^III^–Fe^II^ with a rate constant of k_1_ = 600,000 s^−1^ (Figure 1) [34]. Ru^III^—Fe^II^ is rapidly reduced to Ru^II^—Fe^II^ by the sacrificial donors aniline and 3CP to prevent the back reaction k_2_ [34]. Laser excitation of a 1:1 complex between human Ru-39-Cc and bovine CcO at low ionic strength led to electron transfer from photoreduced heme c to Cu_A_ with rate constants k_3_ = 62,000 ± 10,000 s^−1^ and k_4_ = 30,000 ± 6000 s^−1^, followed by electron transfer from Cu_A_ to heme a with rate constants k_5_ = 20,000 ± 4000 s^−1^ and k_6_ = 2700 ± 600 s^−1^ (Figure 2) (Figure 4). The kinetics of heme c, Cu_A_, and heme a were measured at 550 nm, 830 nm, and 605 nm and fitted to the top line of Figure 2 as previously described [53]. The ratios k_3_/k_4_ and k_5_/k_6_ are consistent with the redox potentials of Cc, CuA and heme a [34,70,71]. There was no change in the rate constants from 0 to 60 mM NaCl. At ionic strengths above 60 mM, the amplitude of the fast intracomplex phase decreased, indicating that the Ru-39-Cc:CcO complex partially dissociated. The resulting uncomplexed Ru-39-Cc reacted with uncomplexed CcO in a new slow phase bimolecular reaction (Figure 5, Figure 2) [34,53]. The fraction of the fast intracomplex phase was f = 0.5 ± 0.1 at 115 mM ionic strength, and the equilibrium dissociation constant K_D_ = 8.0 ± 2 µM was calculated from Equation (3) in the Section 2. The bimolecular phase had a rate constant of k_obs_ = 2400 ± 300 s^−1^. and k_d_ = 2400 ± 400 s^−1^, and k_f_ = 290 ± 40 s^−1^ µM^−1^ were calculated from Equations (4) and (5). At NaCl concentrations above 150 mM, no intracomplex phase was detected, K_D_ > 50 μM, and k_f_ could be determined from Equation (6). k_f_ decreased with increasing ionic strength as expected for a reaction involving electrostatic interactions between the partners. The value of k_d_ was determined at low ionic strength in experiments using 10 μM Ru-39-Cc and 5 μM CcO as described in the experimental procedures section. Under these conditions K_D_ < 0.1 and k_d_ could be determined from Equation (7). For the reaction between human Ru-39-Cc and CcO, k_d_ = 16 s^−1^ at 30 mM NaCl, and increased to 96 s^−1^ at 70 mM NaCl (Figure 5). Horse Ru-39-Cc dissociated from CcO at a lower ionic strength than human Ru-39-Cc, had a larger k_d_ at low ionic strength, and a smaller k_f_ at high ionic strength (Figure 5). These results are consistent with a stronger electrostatic interaction between Human Cc and CcO. Experiments were carried out in 5 mM sodium phosphate pH 7 buffer as well as in 5 mM TrisCl buffer to determine whether the kinetics were sensitive to pH. The ionic strength dependence of k_f_ for the reaction of human Ru-39-Cc with CcO was nearly the same at pH 8.0 (Figure 5) and pH 7.0 (Figure 6). The value of k_f_ was 290 ± 40 µM^−1^ s^−1^ at pH 8.0 and 115 mM ionic strength (Figure 5), compared to k_f_ = 260 ± 40 µM^−1^ s^−1^ at pH 7.0 and 110 mM ionic strength (Figure 6). The ionic strength dependence of k_f_ for the reaction of horse Ru-39-Cc with CcO was also nearly the same at pH 8.0 (Figure 5) as at pH 7.0 [34]. The value of k_d_ was only determined at pH 7.0 (Figure 5 and Figure 6).

### 3.2. Effect of Phosphomimetic Mutations of Human Cc on the Reaction with CcO

The phosphomimetic human Cc mutants T28E, S47E, Y48E, and Y97E were prepared to determine the effect of phosphorylation of Cc T28, S47, Y48, and Y97 on the binding of Cc to CcO and the regulation of the reaction. The effect of the human Cc mutants on the dissociation constant K_D_ of the 1:1 Cc:CcO complex was measured using analytical ultracentrifuge sedimentation velocity experiments of 5 μM Cc and 6 μM CcO in 5 mM sodium phosphate pH 7.0 and 0–300 mM NaCl. At ionic strengths below 60 mM, the K_D_ values of native Cc and all the mutants were <0.2 μM, indicating very strong binding. At ionic strengths greater than 150 mM, the K_D_ values were >30 μM, indicating very weak binding. The K_D_ values of wild-type human Cc and all the phosphomimetic mutants could be measured at an ionic strength of 110 mM (Table 2). The K_D_ values of Cc T28E, S47E, Y48E were all significantly larger than that of wild-type human Cc, indicating that introducing a negative charge at that residue decreases the interaction strength. However, the Cc Y97E did not have a significant effect on K_D_. 

K_d_, k_f_, and k_d_ were measured in 5 mM sodium phosphate, pH 7.0 with 100 mM NaCl. K_d_ was measured by the ultracentrifuge method, and k_f_, and k_d_ were measured by the laser flash photolysis method. The error limits are ±20%.

The kinetics of the reaction between human Cc phosphomimetic mutants and CcO was studied using laser-excited lumiflavin to rapidly transfer an electron to unbound oxidized Cc^o^ to form reduced Cc^r^ [34,69]. Excited lumiflavin cannot transfer an electron to Cc^o^ in the Cc^o^:CcO complex because the heme crevice of Cc is not accessible. Excited lumiflavin also cannot reduce CcO Cu_A_ or heme a [69,72]. The lumiflavin flash photolysis technique is limited to the measurement of rate constants k_obs_ of less than 2000 s^−1^. Flash photolysis of a sample containing 10 μM human Cc, 4.5 μM CcO, 100 μM lumiflavin, 0.5 mM EDTA, 30 mM NaCl in 5 mM sodium phosphate buffer resulted in reduction of unbound Cc, followed by electron transfer from Cc^r^ to CcO with a rate constant of k_obs_ = 13 s^−1^. Since K_D_ < 0.1 μM at low ionic strength, k_d_ can be determined from Equation (7) to be k_d_ = 16 s^−1^ at 30 mM NaCl (Figure 6). As the ionic strength was increased to 100 mM NaCl, k_obs_ increased to 250 s^−1^, and then decreased with further increases in NaCl. At ionic strengths above 150 mM, K_D_ > 30 μM, and k_f_ can be determined from Equation (6) (Figure 6). The T28E and S47E human Cc mutants significantly increased the value of k_d_ at low ionic strength, and decreased the value of k_f_ at high ionic strength (Figure 6, Table 2). The S47E Y48E double mutant decreased the value of k_f_ more than that of the S47E single mutant. These results indicate that phosphorylation of human Cc T28, S47, and Y48 decreases the strength of the interaction between Cc to CcO. 

## 4. Discussion

The analytical ultracentrifuge experiments indicated that the dissociation constant K_D_ = 4.7 μM for the 1:1 complex of human Cc with bovine CcO compared to 28 μM for the complex of horse Cc with bovine CcO complex at 110 mM ionic strength (Table 1). These results are in agreement with earlier studies showing that human Cc binds more strongly to bovine CcO than does horse Cc [72]. Five amino acid residues at the Cc binding domain which differ between horse and human Cc have been identified, 11, 12, 50, 83 and 89 (Figure 3). To explore the role of these amino acid residues in binding CcO, single-point mutations at residues 11, 12, 50, 83 and 89 in horse and human Cc were prepared. It was found that for the human mutants K_D_ decreased in the order: native horse Cc > human A50D > V83A > I11V > native human Cc > M12Q > E89T (Table 1). For the horse mutants K_D_ decreased in the order: T89E > native horse Cc > V11I Cc > Q12M > D50A > A83V > native human (Table 1). The largest effect was observed for the mutants at residue 50, where the horse Cc D50A mutant decreased K_D_ from 28.4 to 11.8 μM, and the human Cc A50D increased K_D_ from 4.7 to 15.7 μM. Cc residue 50 is located towards the back of Cc a distance from the main binding site on CcO subunit II (Figure 1, Figure 2 and Figure 3). However, it is close to subunit 7A, and interactions with this subunit might account for the effect of the mutations (Figure 3). A large effect was also observed for mutations at Cc residue 89, where the horse Cc T89E mutant increased K_D_ to >35, and the human Cc E89T decreased K_D_ to 2.1. Residue 89 is near the CcO subunit II binding domain, and electrostatic repulsion by the negatively charged E89 could explain the effects of these mutants (Figure 3). Horse Cc residue A83 is very close to the interaction domain, and the horse A83V mutation could increase the hydrophobic interaction, accounting for the decrease in K_D_. The horse Q12M and the human M12Q mutations both cause a decrease in K_D_, which might be due to the location of Q12 within the interaction domain and subtle differences in the orientation of horse and human Cc.

Laser flash photolysis studies indicated that horse, human, and human A50D Ru-39-Cc had nearly the same rate constant k_3_ = 60,000 s^−1^ for electron transfer from heme c to Cu_A_ in the Ru-39-Cc:CcO complex at low ionic strength (Figure 5). The complex dissociation rate constant k_d_ for both horse and human Ru-39-Cc increased rapidly with increasing ionic strength, consistent with a strong electrostatic interaction between Cc and CcO (Figure 5). At 40 μM NaCl, k_d_ = 592 for horse Ru-39-Cc, compared to k_d_ = 21 for human Ru-39-Cc, indicating much weaker binding for horse Cc at low ionic strength. As the ionic strength was raised above 120 mM the Cc:CcO complex completely dissociated, and the formation rate constant k_f_ was found to decrease with increasing ionic strength, as expected for an electrostatic interaction. The k_f_ values were smaller for horse and human A50D Ru-39-Cc than for human Ru-39-Cc, indicating weaker binding.

The phosphomimetic human Cc mutants T28E, S47E, Y48E, and Y97E were used to study the effect of phosphorylation of Cc T28, S47, Y48, and Y97 on the binding of Cc to CcO (Figure 6, Table 2). The human Cc mutants T28E, S47E, Y48E all increased the value of K_D_ significantly, indicating weaker binding, while the Y97E mutant had no significant effect on K_D_. The T28E and S47E mutants significantly increased the complex dissociation rate constant k_d_ at low ionic strength, and decreased the complex formation rate constant k_f_ at high ionic strength, consistent with a decrease in the binding strength. Cc T28 is phosphorylated in kidney, causing a 50% decrease in oxygen consumption, and the V_max_ of the reaction of the phosphomimetic mutant Cc T28E with CcO was 78% reduced compared to wild-type Cc [63]. Cc S47 is phosphorylated in brain, causing a 48% decrease in oxygen consumption, and the reaction of the Cc S47E mutant with CcO was decreased by 54% [62]. Cc Y48 is phosphorylated in liver, and the K_m_ of the reaction of the phosphomimetic mutant Cc Y48E with CcO is 3.6-fold larger than the wild-type Cc, indicating weaker binding [73]. Cc Y97 is phosphorylated in heart, and the K_m_ for the reaction with CcO is increased 2-fold [74]. Tyr 97 phosphorylation was also characterized with the replacement of tyrosine with a phosphomimetic, p-carboxymethyl-L-phenylalanine (pCMF), which actually increased the CcO activity [75,76]. Cc is phosphorylated at T58 in rat kidney, which controls mitochondrial respiration and apoptosis [77]. However, isoleucine is present at Cc residue 58 in humans, which lack tissue-specific isoforms such as the testis isoform, and only express a single Cc. The present results are in agreement with previous studies and provide more detailed information on how the phosphomimetic Cc mutants affect the k_f_ and k_d_ for the reaction with CcO.

## Data Availability

Data supporting these results is available in the Ph.D. and Honors Theses of the co-authors in the University of Arkansas Library.

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
