# Peer review of "Accelerated Evolution of Cytochrome c in Higher Primates, and Regulation of the Reaction between Cytochrome c and Cytochrome Oxidase by Phosphorylation"

_cells, 2022, doi:10.3390/cells11244014_

Round 1
Reviewer 1 Report
Authors Brand et al. presented a study on investigation of the functions of several amino acids during the Cc: CcO interaction. Single-point mutants of horse and human Cc were made at each of 5 amino acid changes that occurred from horse Cc to human Cc. The Cc:CcO dissociation constant KD of the horse mutants were investigated and ordered. Besides, the kinetics of the reaction between four human Cc phosphomimetic mutants and CcO was studied using laser-excited lumiflavin to examine the role of Cc phosphorylation of these four amino acid mutations in regulating the reaction of Cc with CcO. The research is very important for understanding functions of key amino acids of Cc in regulation of the reaction with CcO.
My major concern is that authors have four figures and two tables from previous studies. However, authors failed to label these figures and tables in a right way to show if these figures or tables are directly from previous studies (if this is the case, authors should delete these figures and tables) or adopted from previous studies (if this is the case, authors should clearly show they were adopted or summarized from previous studies).
Minors:
Abstract
Cc first shows in the manuscript should be listed the full name followed by the abbreviation. Please change the first “Cytochrome” to “Cytochrome c (Cc)”.
Line 44, please change “Rb. Sphaeroides” to “Rhodobacter sphaeroides”
Figure 1. Is this bacterium Cc2 or horse Cc or human Cc? Please show the PDB number or provide citations in the figure legend. If this figure has been published previously, please give the credit to the paper. If this figure was generated based on previous studies, please add “adopted from”
Figure 2. Similarly, please provide the citation in the figure legend.
Line 99, Rb. Sphaeroides should be italicized
Lines 84 and 109, one shows “cytochrome oxidase (CcO)” the other shows “cytochrome c oxidase (Cco)”, which is confused! Please clarify and keep it consistent in the context of MS.
Line 130, change “Rhodobacter sphaeroides” to “R. Sphaeroides”
Table 1. The title of this table should be put on the top of the table. And add reference 53 in the title since this study had been published. Is th.is data from reference 53 directly or after summarization? If it was directly from reference 53, the table 1 should be deleted. If not, please clarify in the title by adding “adopted from reference 53”.
Line 158. Delete “Data from (53)”.
Line 407 and 408, what is this star “*” here for?
Author Response
Thank you for your detailed review and valuable suggestions for revision. We have revised the manuscript according to your suggestions as follows:
Question 1:
My major concern is that authors have four figures and two tables from previous studies. However, authors failed to label these figures and tables in a right way to show if these figures or tables are directly from previous studies (if this is the case, authors should delete these figures and tables) or adopted from previous studies (if this is the case, authors should clearly show they were adopted or summarized from previous studies).
Answer 1:
In response to the reviewers’ suggestions, we have removed the original Figure 2 (Line 109) and the original Table 1 (line 171) and Table 2 (line 176) from the manuscript that summarized previous work in the Introduction. We have revised Figure 1 (line 63) to show all the amino acids of cytochrome c that we mutated to determine the effect of evolution from horse to human Cc, a major goal of this paper. This figure is based on a crystal structure of horse Cc in the protein data bank, which we reference in the caption, and has not been published previously showing these amino acids. The original Figure 3 (Now Figure 2) (line 163) has been revised to show all the amino acids that we have mutated to study both the evolution of Cc and the effect of phosphorylation of Cc amino acids (line 163). The crystal structures it is based on are given in the caption. It has not been previously published showing these amino acids. The original Figure 4 (Now Figure 3) (line 234) is the same as in the original manuscript. It has not been previously published showing these amino acids.
Question 2:
Minors:
Abstract: Cc first shows in the manuscript should be listed the full name followed by the abbreviation. Please change the first “Cytochrome” to “Cytochrome c (Cc)”.
Line 44, please change “Rb. Sphaeroides” to “Rhodobacter sphaeroides”. Etc.
Answer 2:
We have addressed all of the minor points raised by the reviewer. The abbreviations and italics have been revised throughout the manuscript to be consistent. The * in lines 407 and 408 of the original manuscript has been removed. It referred to the excited state of lumiflavin.
All the other minor points were about the Figures and Tables that have been removed or revised, as discussed in Question 1 above.

Reviewer 2 Report
The manuscript by Brand and colleagues studies the role of specific amino acid replacements found in human versus horse cytochrome c. In addition, the authors study the effect amino acids that have been shown to be phosphorylated and affect the steady state kinetics of cytochrome c and CcO. The authors are experts in fast kinetics cytochrome c and CcO using the flash photolysis approach in which a ruthenium complex reduces cytochrome c upon laser activation. The manuscript is well presented and I have only minor points to further polish the manuscript.
-Please make a mention of what the effect of adding the ruthenium group has on the kinetics with complexes III and IV (for example, are the steady state kinetics altered?).
-Since the study uses horse and human cytochrome c and bovine cytochrome c oxidase it should be mentioned how similar human/horse cytochrome c are in comparison to the bovine protein. Are any amino acids different in bovine cytochrome c that are studied in the manuscript?
-Please justify the use of pH 8 (more similar to matrix pH) instead of pH 7.4 (more similar to intermembrane space pH where the reaction takes place) for the experiments.
-The statement “Tyr 97 phosphorylation was also characterized with the replacement of tyrosine with a more realistic phosphomimetic, p-carbox-ymethyl-L-phenylalanine (pCMF), which actually increased the CcO activity (75,76)” is confusing because this “more realistic” phosphomimetic shows the opposite effect as cytochrome c phosphorylated on this site, so it’s clearly not a more realistic phosphomimetic. The sentence could simply be taken out; if the authors wish to include this reference they should clarify that this is not a more realistic phosphomimetic replacement.
-One important issue is not discussed but is crucial in this context: the loss of the testes-specific isoform of cytochrome c in humans and that the current version of the human protein is more or less a mix of the somatic and testes isoforms of other species. This could be added to the discussion section or as part of the paragraph where cytochrome c/CcO coevolution is discussed.
Some minor editing is needed including:
- the “c” in cytochrome c and “bc” in bc1 should be in italics throughout
-line 77: with a rate constant
- some the Angstrom symbols should be fixed
-Figure 3 legend: assign colors (orange and grey) to COX subunits
Author Response
Thank you for your detailed review and valuable suggestions for revision. We have revised the manuscript according to your suggestions as follows:
Question 1:
Please make a mention of what the effect of adding the ruthenium group has on the kinetics with complexes III and IV (for example, are the steady state kinetics altered?).
Answer 1:
In response to the reviewers’ comments, we have added a sentence stating that the ruthenium is on the back side of Ru-39-Cc and previous studies have shown that it has the same steady-state kinetics as wild-type Cc (lines 130-132 in revised manuscript).
Question 2:
Since the study uses horse and human cytochrome c and bovine cytochrome c oxidase it should be mentioned how similar human/horse cytochrome c are in comparison to the bovine protein. Are any amino acids different in bovine cytochrome c that are studied in the manuscript?
Answer 2:
We added a sentence stating that there only 2 amino acid substitutions between horse and bovine Cc, which should not have a significant effect on the kinetics (lines 214 to 217).
Question 3:
Please justify the use of pH 8 (more similar to matrix pH) instead of pH 7.4 (more similar to intermembrane space pH where the reaction takes place) for the experiments.
Answer 3:
We added several sentences explaining that we carried out kinetic studies at both pH 7 and pH 8 to study the effect of pH. There was no significant difference in the kinetics between pH 7 and pH 8 (lines 426 -433).
Question 4:
The statement “Tyr 97 phosphorylation was also characterized with the replacement of tyrosine with a more realistic phosphomimetic, p-carboxymethyl-L-phenylalanine (pCMF), which actually increased the CcO activity (75,76)” is confusing because this “more realistic” phosphomimetic shows the opposite effect as cytochrome c phosphorylated on this site, so it’s clearly not a more realistic phosphomimetic. The sentence could simply be taken out; if the authors wish to include this reference they should clarify that this is not a more realistic phosphomimetic replacement.
Answer 4:
We revised the sentence about the phosphomimetic Tyr modification on lines 545 – 548.
Question 5:
One important issue is not discussed but is crucial in this context: the loss of the testes-specific isoform of cytochrome c in humans and that the current version of the human protein is more or less a mix of the somatic and testes isoforms of other species. This could be added to the discussion section or as part of the paragraph where cytochrome c/CcO coevolution is discussed.
Answer 5:
We added a brief discussion of the fact that there are no tissue specific isoforms such as the testis isoform in humans (lines 548-550).
Question 6:
Some minor editing is needed including:
- the “c” in cytochrome c and “bc” in bc1 should be in italics throughout
-line 77: with a rate constant
- some the Angstrom symbols should be fixed
-Figure 3 legend: assign colors (orange and grey) to COX subunits
Answer 6:
The “c” in cytochrome c and Cc and the “bc” in bc1 will be changed to italics throughout the manuscript.
I think that the current “with rate constant keta = 2 x 106 s-1 “ is OK grammatically.
The Angstrom symbols have been corrected throughout the manuscript.
The Figure 2 legend (former Figure 3) has been revised as suggested (lines 163 -169).

Reviewer 3 Report
44. pp. Rb. Sphaeroides corrected: Rhodobacter sphaeroides
50. pp. Sphaeroides corrected: sphaeroides
60-62. pp. In the Figure 1., is R. sphaeroides cytochrome c shown? Mark it!
Human and horse cytochrome c differ in 11 amino acids (BLAST). These should be summarized in a table and discussed. With special regard to the differences that change the charge (D51A, K61G, E90T, A93E; human-horse).
The format of the references is not uniform. It needs to be fixed.
Author Response
Thank you for your detailed review and valuable suggestions for revision. We have revised the manuscript according to your suggestions as follows:
Question 1:
- pp. Rb. Sphaeroides corrected: Rhodobacter sphaeroides
- pp. Sphaeroides corrected: sphaeroides
Answer 1:
In response to the reviewers’ comments, the format of Rb. Sphaeroides has been standardized throughout the manuscript.
Question 2:
60-62. pp. In the Figure 1., is R. sphaeroides cytochrome c shown? Mark it!
Answer 2:
The Figure 1 caption has been revised to indicate that it is the structure of horse Cc, and the Protein Data Bank reference is given.
Question 3:
Human and horse cytochrome c differ in 11 amino acids (BLAST). These should be summarized in a table and discussed. With special regard to the differences that change the charge (D51A, K61G, E90T, A93E; human-horse).
Answer 3:
A section has been added listing all the amino acid mutations from horse Cc to human Cc (lines 211 – 217). There are tables of these mutations in the references given. Table 1 lists the mutations studied in this manuscript. It should be noted that in a Blast search, the horse and human Cc are listed with 105 amino acids, including methionine at the amino terminus. However, this initial formyl-methionine is removed by posttranslational modification, and the Glycine at position 1 is acetylated. The mature horse and human Cc have 104 amino acid residues starting with acetylated Gly and the numbering used in the manuscript is based on this.
Question 4:
The format of the references is not uniform. It needs to be fixed.
Answer 4:
I realize that some of the references are in a different format. I assume that your journal changes the format of the references to a common format. Please let me know if I should standardize the reference format.
To Alize Li:
Question:
We just sent you a revision request. Please note that some sentences highlighted in the attachment are similar to part of other published papers. In case of any unnecessary trouble, we kindly ask you to depict this part in another way. Please see the attachment file with the green highlight section.
Answer:
Thank you for using the iThenticate report for finding the repetitions in the current manuscript from our previous manuscripts. These were primarily in the standard procedures section where we use some of the same methods for different studies. I have gone through the iThenticat report and removed all the repetitions shown in green.
These revisions have been made at the following lines in the revised manuscript:
Lines 108 – 112: This old Figure 2 and its caption has been removed in the revised manuscript.
Lines 153 – 156: This sentence has been removed and replaced with the sentence on lines 156 – 158.
Lines 163 – 169: The Figure 2 Caption (Old Figure 3) has been revised to remove duplication.
Lines 170 – 185: Tables 1 and 2 have been removed completely.
Lines 301 – 317: These sentences have been removed and replaced with the sentences on lines 298 -300 and 317 – 321 to avoid duplication.
Lines 385 – 387: These lines have been removed and replaced to avoid duplication.
Lines 394 – 402: These lines have been removed and replaced to avoid duplication.
Lines 411 – 414: These lines have been removed and replaced to avoid duplication.
